# Deep Proteomic Investigation of Metabolic Adaptation in Mycobacteria under Different Growth Conditions

**DOI:** 10.3390/proteomes11040039

**Published:** 2023-12-07

**Authors:** Mariia Zmyslia, Klemens Fröhlich, Trinh Dao, Alexander Schmidt, Claudia Jessen-Trefzer

**Affiliations:** 1Faculty of Chemistry and Pharmacy, University of Freiburg, Albertstrasse 21, 79104 Freiburg, Germany; mariia.zmyslia@ocbc.uni-freiburg.de (M.Z.); trinh.dao@cibss.uni-freiburg.de (T.D.); 2Proteomics Core Facility, Biozentrum Basel, University of Basel, Spitalstrasse 41, 4056 Basel, Switzerland; klemens.froehlich@unibas.ch (K.F.); alex.schmidt@unibas.ch (A.S.); 3Spemann Graduate School of Biology and Medicine (SGBM), University of Freiburg, Albertstrasse 19A, 79104 Freiburg, Germany; 4The Center for Integrative Biological Signaling Studies, University of Freiburg, 79104 Freiburg, Germany

**Keywords:** mycobacteria, proteomics, carbon source, metabolism

## Abstract

Understanding the complex mechanisms of mycobacterial pathophysiology and adaptive responses presents challenges that can hinder drug development. However, employing physiologically relevant conditions, such as those found in human macrophages or simulating physiological growth conditions, holds promise for more effective drug screening. A valuable tool in this pursuit is proteomics, which allows for a comprehensive analysis of adaptive responses. In our study, we focused on *Mycobacterium smegmatis*, a model organism closely related to the pathogenic *Mycobacterium tuberculosis*, to investigate the impact of various carbon sources on mycobacterial growth. To facilitate this research, we developed a cost-effective, straightforward, and high-quality pipeline for proteome analysis and compared six different carbon source conditions. Additionally, we have created an online tool to present and analyze our data, making it easily accessible to the community. This user-friendly platform allows researchers and interested parties to explore and interpret the results effectively. Our findings shed light on mycobacterial adaptive physiology and present potential targets for drug development, contributing to the fight against tuberculosis.

## 1. Introduction

Tuberculosis (TB) remains a major cause of morbidity and mortality worldwide, claiming an estimated 1.6 million lives each year [1]. About a quarter of the world’s population is infected with *Mycobacterium tuberculosis* (*M. tuberculosis*, *Mtb*), the causative agent of TB. Despite the reported success of directly observed short-term therapy (DOTS), lack of treatment adherence is one of the factors leading to the emergence of multidrug-resistant strains (MDR-TB). MDR-TB refers to strains that are resistant to two or more of the four first-line tuberculosis drugs (isoniazid, rifampin, pyrazinamide, and ethambutol). Patients infected with MDR-TB strains have a mortality rate of 60–90%, which is similar to patients who do not receive treatment. This is exacerbated by the rapid progression of the disease and the increased risk of reactivation of latent TB in people infected with the human immunodeficiency virus (HIV). Given the increasing burden of tuberculosis worldwide, there is an urgent need to find new therapeutics that are effective against wild-type and MDR strains and have a better sterilizing effect. Moreover, there has been a steady global increase in the occurrence and fatality rates associated with both fast and slow-growing non-tuberculous mycobacterial (NTM) diseases [2].

An incomplete understanding of the mechanisms associated with mycobacterial pathophysiology, signaling pathways, and the adaptive response to intracellular phagocytosis remain major obstacles to significant progress in drug development [3,4]. The uncertainties in the biology of tuberculosis disease greatly complicate efforts in developing novel treatments. Rational approaches to drug development have not led to the desired success in recent years and have not yet yielded drugs that have progressed to clinical testing. Successful campaigns were founded on phenotypic whole-cell screening, followed by subsequent target identification [5]. One lesson learned during these screening campaigns is the importance of the screening conditions used and how well they resemble human disease. Several examples in recent drug development campaigns underscore the importance of screening under physiologically relevant conditions in vitro to avoid unnecessary and costly drug discovery efforts [6,7]. For example, studies on the mechanism of action of pyrimidine imidazoles revealed that the whole-cell activity of the analogs depends on the presence of glycerol in the growth medium [8]. The mechanism of resistance indicated the inactivation of a mycobacterial glycerol kinase, which meant that glycerol could not be degraded as a carbon source. However, the lacking phenotype of an *Mtb* glycerol kinase mutant phenotype in a mouse model indicated that glycerol metabolism in mice is unlikely to be a susceptible pathway. The dependence of *Mtb* on glycerol metabolism during human infection remains unknown, and hence it is uncertain if a further development of this compound class is recommended. One can conclude from this and similar observations that high-content screening for the discovery of antimycobacterial compounds in primary human macrophages infected with *M. tuberculosis* would be a reasonable approach for directly targeting mycobacteria in a “host-like” environment [9,10]. Another, simpler possibility would be to screen mycobacteria in classical liquid cultures mimicking physiological growth conditions, e.g., at different stages of infection. Mycobacteria regulate their metabolism in response to environmental factors that occur during infection, such as acidic pH, actual availability of host-associated carbon sources or defense molecules, and oxygen limitation, to name only some examples [11]. The response to the presence of host-associated carbon sources is believed to play a fundamental role in mycobacterial adaptive physiology [12]. Analyzing this response at the protein level, and hence the (conditionally) present drug targets, is key to understanding and implementing physiological growth conditions in the culture flask.

In the last decade, proteomics analysis has been applied as a tool for the global assessment of changes at the cellular protein level, complementing genomic and metabolomics analysis pipelines.

In this study, we used a straightforward, solution-based, label-free method for qualitative and quantitative proteomic analysis using LC-MS. Traditionally, quantitative proteomics has relied on stable isotope labeling and these labeling strategies provide elementary alternatives to former 2DE-MS; however, they require several sample preparation steps and are costly [13,14]. Moreover, the number of samples that can be analyzed in parallel is limited. Gel-based methods or pre-fractionation have also been applied to reduce sample complexity [15]. Additionally, classical analysis methods aim to identify a maximum number of peptides rather than proteins. While this approach is completely valid for many applications, the investigation of different growth conditions in a pathogen would greatly benefit from focusing on proteome coverage rather than individual protein coverage. We therefore employed a recent measurement method that focuses on peptide ions present in a small, but highly populated, mass-to-charge window. It has previously been shown that this increases proteome coverage [16].

Given the well-defined biochemical data and supporting genomic data, we chose to study the protein expression profile of the model organism *Mycobacterium smegmatis* (*M. smegmatis*) mc^2^155, grown under various culture conditions. While *M. smegmatis* may not be a comprehensive representation of pathogenic species like *Mtb*, the method developed holds promising potential for future applications in studying pathogenic species. Our primary interest is in understanding the relative effects of host-associated carbon sources on mycobacteria; the methods used in this study are applicable to the investigation of a wide range of adaptive responses. In addition, with our data set, we have created an online tool available to the community, in which individual proteins or conditions can be searched for (https://klemens-froehlich.shinyapps.io/Mycobacterium/) (accessed on 4 December 2023).

Protein abundance is heavily influenced by mycobacterial culture conditions. However, proteome-wide dimensions of a simple exchange of the applied carbon source during bacterial culture growth have never been assessed systematically in a high-throughput manner. We developed a straightforward and cheap protocol to perform whole-proteome analysis of several sample sets in parallel without any isotope-labeling strategy required. Since mycobacterial adaptive physiology is influenced by the presence of available carbon and energy sources in the environment, we chose to compare six different conditions applying various carbon sources individually. We employed cholesterol, D-glucose, L-lactate, glycerol (50 mM), as well as Tween 80 (0.4 µM) in minimal medium and added a “complex-culture-medium condition” containing several carbon and energy sources (Dulbecco’s Modified Eagle Medium with 10% fetal calf serum, DMEM, and 10% FCS). All samples were prepared in quadruplicates. Cholesterol, L-lactate, and D-glucose were chosen, as these molecules have been reported as host-associated carbon sources available to the engulfed mycobacterium during different stages of infection. D-glucose is a primary carbon source at the onset of infection [17]. At later stages, *Mtb* stimulates a robust shift towards aerobic glycolysis in macrophages, leading to a pro-inflammatory effector function in conjunction with increased production and secretion of L-lactate [18]. As a result, L-lactate is highly abundant at the infection site [19,20]. *Mtb* can utilize L-lactate as a sole carbon source for in vitro growth and a ΔlldD2 mutant, lacking one of two L-lactate dehydrogenases, is impaired in replication in human macrophages, indicating a critical role for L-lactate oxidation during intracellular growth. Another important host-associated carbon source is cholesterol. In recent years, several groups have investigated metabolic and transcriptional changes induced by cholesterol, since fatty acids and cholesterol are proposed to act as major carbon and energy sources for *Mtb* in vivo, especially at later stages of infection [21,22,23]. Glycerol and Tween 80 were each applied as both molecules are traditionally used in mycobacterial culture and drug screening assays to promote growth. DMEM contains D-glucose (40 mM), sodium pyruvate (1 mM), *myo*-inositol (0.04 mM), calcium pantothenate (8.4 µM), and various amino acids as potential energy sources and is often applied in mammalian cell culture, e.g., to cultivate macrophages during an infection experiment. We decided to include DMEM in our set of conditions since it challenges the analysis method due to the high foreign protein background (FCS).

## 2. Experimental Procedures

### 2.1. Bacterial Strains and Cell Culture

*M. smegmatis* strain mc^2^155 was kindly provided by William R. Jacobs Jr. (Albert Einstein College of Medicine, New York, NY, USA). Routinely, *M. smegmatis* strain mc^2^155 was grown in Middlebrook 7H9 supplemented with 10% (*v*/*v*) OADC (0.05% oleic acid, 5% albumin, 2% dextrose, 0.004% catalase, 0.85% NaCl) and 0.05% (*v*/*v*) Tween 80.

Prior to the experiment, cells were pre-cultured as described above, washed three times with PBS, and thoroughly resuspended to determine OD_600_.

All six conditions were performed in quadruplicate as described below. For five conditions, minimal Sauton’s medium (0.5 g/L monobasic potassium phosphate, 0.5 g/L magnesium sulfate, 4.0 g/L L-asparagine, 2.0 g/L citric acid, 0.05 g/L ferric ammonium citrate, and 0.1 mL/L 1% zinc sulfate solution, pH 7.0) was supplemented with the following individual carbon sources: glycerol, D-glucose, magnesium L-lactate, cholesterol, or Tween 80 to final concentrations of 50 mM, 50 mM, 25 mM (equivalent to 50 mM L-lactate), 1.3 mM, and 0.34 mM, respectively. High-glucose DMEM supplemented with 10% FCS was used as the last “complex culture medium condition” containing several carbon and energy sources. The prepared medium was inoculated with washed pre-culture to a starting OD_600_ of 0.01.

Cells were grown until the OD_600_ reached 0.6–0.8 (see Appendix A for growth curves). Subsequently, cells were harvested by centrifugation (5000× *g* for 5 min at 4 °C) and washed in PBS, and the pellet was quickly frozen in liquid nitrogen and stored at −80 °C until further use.

### 2.2. Protein Extraction

Cell pellets were thawed on ice, resuspended in 0.5 mL of 5% SDS in 1× PBS (*w*/*v*), and lysed by sonication on ice using Bandelin Sonopuls HD 2070 ultrasonic homogenizer (Bandelin Electronics, Berlin, Germany, microtip of 3 mm in diameter) for 30 s on/off 2 times at 50% amplitude. The cleared lysate was collected by centrifugation at 14,000× *g* for 10 min at 4 °C. The total protein concentration of the lysate was determined by measuring tryptophan fluorescence.

Following lysis, 20 µg of cell lysate was subjected to digestion with the SP3 approach [24] using a Freedom Evo 100 liquid handling platform (Tecan Group Ltd., Männedorf, Switzerland). In brief, Speed Beads^TM^ (#45152105050250 and #65152105050250, GE Healthcare, Chicago, IL, USA) were mixed 1:1, rinsed with water, and diluted to make an 8 μg/µL stock solution. Samples were adjusted to a final volume of 90 µL and 10 µL of the bead stock solution was added to them. Proteins were bound to the beads by the addition of 100 µL of 100% acetonitrile to the samples, which were then incubated for 8 min at RT with gentle agitation (200 rpm). Afterward, samples were placed on a magnetic rack and incubated for 5 min. Supernatants were removed and discarded. The beads were washed twice with 160 µL of 70% (*v*/*v*) ethanol and once with 160 µL of 100% acetonitrile. Samples were taken off the magnetic rack and 50 µL of digestion mix (10 ng/µL of trypsin in 50 mM triethylammonium bicarbonate) was added to them. Digestion was allowed to proceed for 12 h at 37 °C. After digestion samples were placed back on the magnetic rack and incubated for 5 min, supernatants containing peptides were collected and dried under a vacuum.

### 2.3. Reversed-Phase Liquid Chromatographic and Mass Spectrometry Analysis

For each sample, 0.1 µg total peptides were subjected to LC-MS analysis on an Orbitrap Exploris 480 mass spectrometer equipped with a nanoelectrospray ion source (both Thermo Fisher Scientific, Waltham, MA, USA). Peptide separation was carried out using a Neo Vanquish system (Thermo Fisher Scientific) equipped with an RP-HPLC column (75 μm × 30 cm) packed in-house with C18 resin (ReproSil-Pur C18-AQ, 1.9 μm resin; Dr. Maisch GmbH, Ammerbuch, Germany) and a custom-made column heater (60 °C). Peptides were separated using a linear gradient of 96% solvent A (0.1% formic acid, 99.9% water) and 4% solvent B (80% acetonitrile, 0.1% formic acid, 19.9% water) to 10% solvent B over 10 min, further to 35% solvent B over 30 min and 50% solvent B over 6.5 min at a flow rate of 0.2 µL/min.

For data-independent acquisition (DIA) analysis, each MS1 scan was followed by high-collision dissociation (HCD) of 60 isolation windows covering the mass range between 498 and 742 with 4 *m*/*z* width and no overlap. For MS1, 300% AGC ions were accumulated in the Orbitrap cell over a maximum time of 22 ms and scanned at a resolution of 15,000 FWHM (at 200 *m*/*z*). To maximize the dynamic range and the sensitivity of the method, an MS2 AGC target of 3000% was used. The accumulation time was set to 55 ms to limit the cycle time. A resolution of 30.000 (at 200 *m*/*z*) was used. The stepped collision energy was set to 22, 26, and 30% and one microscan was acquired for each spectrum. Acquired raw files were converted to mzml format using MSConvert [25].

### 2.4. Bioinformatics Analysis

For DIA data analysis, DIA-NN 1.8 [26] was employed with the recommended settings. In brief, a FASTA database containing the *M. smegmatis* mc^2^155 proteome was obtained from Uniprot containing 6559 proteins. A spectral library was predicted covering the mass range between 490 and 750 *m*/*z*, using trypsin specificity and charge states between 1 and 4. Carbamidomethylation (C) was set as a fixed modification. The match between runs option was activated for the main analysis.

Downstream analysis was performed using R 4.2.2 employing the following packages: tidyverse [27], limma [28], ComplexHeatmap [29], and ClusterProfiler [30]. The subcellular localization of proteins was predicted using STRING’s “Annotate your Proteome” function [31].

Our enrichment analysis is based on predicted terms since the experimental characterization of *M. smegmatis* is very limited.

### 2.5. Implementation of the Shiny App

The output of limma (based on the DIA-NN protein group matrix) and the PCA results were each merged with the string pathway predictions as input files for the shiny app. The visualization can be filtered for any growth comparison compared against (i) glycerol or (ii) all other conditions combined. Additionally, a protein-level PCA can be visualized to investigate the relationship of enrichment terms in terms of their individual protein abundance across conditions. It has to be noted that protein grouping may lead to duplicate displays of protein abundances in very rare cases in the online shiny app. In this case, we recommend downloading the protein group matrices from MASSIVE to investigate more complex protein groupings.

### 2.6. Data Availability

All raw files, file annotation, FASTA database, DIA-NN output, STRING prediction output, and the R code have been deposited at MASSIVE and are accessible via the identifier MSV000092025 (password “myco”). The code for the Shiny app is available upon reasonable request from the authors.

### 2.7. RNA Extraction and Reverse-Transcription PCR

Cultures were cultivated until the OD_600_ reached a range of 0.6–0.8. Subsequently, cells were harvested via centrifugation, and the resulting cell pellets were promptly frozen in liquid nitrogen before storage at −80 °C until further utilization. Total RNA extraction was carried out using the RNeasy Mini-Kit (Qiagen, Hilden, Germany) in accordance with the manufacturer’s guidelines. The concentration and quality of the extracted RNA were determined spectrophotometrically, while the integrity of each RNA sample was verified through agarose gel electrophoresis. The isolated RNA was then stored at −80 °C in nuclease-free water.

For RT-PCR analysis, the primer pairs specified in Appendix A were employed, and the OneTaq One-Step RT-PCR Kit (New England Biolabs, Ipswich, MA, USA) was used following the supplier’s provided protocol. MSMEI_2690 primers served as the control in the analysis. Analysis was performed in the presence or absence of reverse transcriptase.

## 3. Results and Discussion

*M. smegmatis* wild-type pre-cultures were incubated in a classical 7H9 medium. After overnight incubation, pre-cultures were washed thrice with phosphate-buffered saline (PBS) and used to inoculate either minimal medium with the respective carbon source or DMEM, reaching the logarithmic phase (Figure 1). To avoid interference, each component was freshly purchased at analytical grade. Glycerol and D-glucose were dissolved at 20× concentration, while L-lactate and Tween 80 were dissolved at 8× and 500× concentrations, respectively, in minimal medium and were filter-sterilized. Cholesterol, challenging to dissolve in water-based buffers, was aseptically weighed, applied directly at the final concentration in minimal medium, and stirred for 1 h. Unlike prior methods using tyloxapol or cyclodextrin, we did not solubilize cholesterol to prevent potential interference from additives serving as energy sources. Bacterial cell pellets were collected by centrifugation, washed with PBS, and whole-cell proteins were extracted in sample buffer with 5% SDS. Unlike our previous protocol, where we pre-fractionated samples with Triton X-100 for lipoprotein identification, we modified the extraction buffer to isolate the whole proteome. An automated SP3 protocol processed all extracts in parallel, followed by LC-MS/MS for label-free analysis.

### 3.1. Sample Complexity, Ion Detection, and Protein Identification

Proteome analysis was carried out using SWATH, which allows for accurate and precise proteome profiling [32].

The number of identified proteins can be seen in Figure 2A. A total of 4230, 4193, 4318, 4124, 4464, and 4203 proteins were identified on average for cholesterol, DMEM, D-glucose, glycerol, L-lactate, and Tween 80, respectively. Interestingly, the growth condition “L-lactate” showed the highest number of proteins. Earlier findings by Serafini et al., applying stable isotope labeling, revealed that *M. tuberculosis* exhibits a remarkable adaptation to utilize both L-lactate and pyruvate as energy sources [14]. This metabolic process involved gluconeogenesis, valine metabolism, the Krebs cycle, the GABA (*γ*-aminobutyric acid) shunt, the glyoxylate shunt, and the methylcitrate cycle. Notably, the methylcitrate cycle and the glyoxylate shunt, traditionally associated with fatty acid metabolism, unexpectedly operated in reverse in *M. tuberculosis*, facilitating optimal utilization of L-lactate and pyruvate. Hence, the high number of proteins found in our study in *M. smegmatis*, in the L-lactate condition, potentially reflects the utilization of multiple pathways leading to an increased number of produced proteins. Visualization of missing values at the protein group level has been performed and is shown in Appendix A.

Quantitative precision was measured by calculating the coefficients of variation (CVs) of non-logarithmic protein group abundances (Figure 2B).

The number of identified precursors can be seen in Figure 2C. A total of 28,451, 28,538, 30,920, 27,063, 32,982, and 29,376 protein groups were identified on average for cholesterol, DMEM, D-glucose, glycerol, L-lactate, and Tween 80, respectively. As expected, the lactate condition led to the highest number of identified precursors, while also displaying the lowest CV distribution (Figure 2D).

### 3.2. Data Quality Assessment and Clustering

To evaluate quantification data quality, we initially assessed the clustering of individual samples and conditions. Figure 3A presents a hierarchical clustering analysis at the protein group level for all six conditions, revealing distinct clusters. Cholesterol and Tween 80 form a separate cluster, distinct from other growth media conditions. DMEM forms a subcluster within a larger cluster of four conditions. This observation is explained by the catabolism of cholesterol and Tween 80 (IUPAC name: Polyoxyethylene (80) sorbitan monooleate), both derived from long-chain fatty acids. While cholesterol degradation is extensively studied in mycobacteria, Tween 80 utilization is less understood, to be detailed in the comparative analysis section. D-glucose, L-lactate, and glycerol, being carbohydrates or their derivatives, likely cluster due to overlapping metabolic pathways. DMEM, a complex medium with various nutrients, stands out due to its unique composition. Mycobacteria’s ability to co-catabolize different carbon sources suggests multiple active metabolic pathways under these growth conditions. Principal Component Analysis (PCA, Figure 3B) was conducted to visualize variance between replicates and reduce dataset dimensionality. The first principal component (explaining 48% of variance) visually separates cholesterol and Tween 80 from other conditions. The second principal component (explaining 17% of variance) distinguishes DMEM from the remaining growth conditions.

Both the hierarchical clustering analysis and the PCA show a close relation between replicates while clearly separating the different growth conditions from each other, showcasing the high quality of the dataset.

### 3.3. Quantitative Analysis

A total of 4569 protein groups were identified across all conditions and samples, spanning approximately five orders of magnitude. This accounts for 68% of predicted genes (6746 protein groups post DIA-NN spectral library generation), showcasing the very good coverage of our analysis method. Mycobacterial proteomes, known for their complexity, play a crucial role in the bacteria’s survival within host cells and evasion of immune responses. This complexity arises from specialized proteins in lipid metabolism, cell wall synthesis, and virulence factors. It is essential to note that not all cellular proteins are produced under the tested conditions. Our identified protein set encompasses 1568 out of 2079 (75%) predicted membrane-associated proteins (GO:0016020), 1919 out of 2485 (77%) cytoplasmic proteins (GO:0005737), and 186 out of 265 (70%) periplasmic/extracellular proteins (GO:0005576). This indicates comprehensive coverage across all cellular compartments. Notably, achieving a 75% coverage of membrane-associated proteins is particularly promising and a distinctive advantage of our extraction procedure. (Mycobacterial) membrane-associated proteins are notoriously challenging to solubilize and identify in complex protein mixtures. Given their significance in cellular processes, disease involvement, and potential as therapeutic targets, enhancing the study of these proteins is critical [33].

### 3.4. Comparative Analysis

Next, we performed a differential expression analysis protocol and compared all samples against the glycerol condition. Glycerol has long been known to be the preferred source of energy and carbon for most mycobacterial species under in vitro conditions, but it does not play a major role under physiological growth conditions. Firstly, we performed limma analysis and generated volcano plots of each differential expression analysis (Figure 4) and further analyzed identified proteins according to their annotated cellular process (Figure 5). Interestingly, we found that D-glucose, cholesterol, and Tween 80 showed distinct enrichment profiles in comparison to glycerol, while DMEM and L-lactate displayed only minor differences to the glycerol condition in the cellular pathway analysis. This is a very interesting finding since DMEM contains multiple carbon, protein, and energy sources and we would have expected a differential regulation compared to glycerol in many pathways. L-lactate, however, is converted to pyruvate or directly to acetate and readily feeds into the Krebs cycle. Glycerol is converted to glycerol-phosphate (aerobic) and from there converted to pyruvate; hence, similar metabolic profiles of these two nutrients are readily explained.

### 3.5. Investigation of L-Lactate as Energy Source

One of the top hits in the differential expression analysis is MSMEI_3870 (I7FNP4), an L-lactate 2-monooxygenase, and we hypothesize that its metabolism to acetate plays a major role in the applied conditions [34]. As a control, we verified the differential expression of lactate monooxygenase in all tested conditions using reverse-transcription PCR (Appendix A). Interestingly, we have further identified MSMEI_5033 (I7G6Z3), an uncharacterized transporter belonging to the sodium solute transporter superfamily, among the top upregulated hits. This class of proteins is involved in the sodium- or proton-coupled solute uptake of diverse small molecule substrates such as acetate, glucose, pyruvate, or lactate [35]. Lactate transport is very well studied in eukaryotic cells and a validated drug target [36]; however until now, the transporter responsible for uptake in mycobacteria has not been identified. We propose from our data that MSMEI_5033 and the homolog protein Rv3728 from *M. tuberculosis* are potentially involved in lactate uptake in mycobacteria.

Interestingly, several of the top downregulated hits (>three-fold log_FC_) are involved in mycobactin synthesis. We identified four homologs of phenyloxazoline synthases MSMEI_3627 (I7FF39), MSMEI_4403 (I7G5F3), MSMEI_4399 (I7FQ79), and MSMEI_4398 (I7G5E0, MbtF) (Figure 6 and Appendix A). Mycobactin is a mycobacterial siderophore responsible for the complexation of iron to shuttle free extracellular iron ions into the cytoplasm of mycobacterial cells. Our data indicate that in the presence of L-lactate, which is a known iron complexing agent, mycobactin is less necessary and therefore genes required for its synthesis are downregulated. Two more proteins of the mycobactin synthesis pathway were identified to be downregulated within the protein set: the L-lysine N6-monooxygenase MSMEI_4397 (two-fold log_FC_, I7GCE0, MbtG) and the fatty-acid-[acyl-carrier-protein] ligase MSMEI_2082 (two-fold log_FC_, A0QUA1, FaD33), further confirming the non-essential mycobactin synthesis in the presence of lactate.

We further investigated the requirement for siderophore synthesis in all other conditions (D-glucose, DMEM, cholesterol, and Tween 80), and we noticed that the glycerol condition clearly stood out, showing increased production of mycobactin synthesis-related proteins (up to four-fold) compared to all other tested conditions (Appendix A). This finding is in line with a previous report, investigating the occurrence of carboxymycobactin in cultures from *M. smegmatis* grown on glycerol or D-glucose. The amount of carboxymycobactin was about 20 times higher when cultures were grown with glycerol instead of D-glucose [37]. Siderophores play a well-documented role in pathogen virulence, and mycobactin is essential for both the virulence and survival of intracellular *M. tuberculosis*. Siderophore-inspired design of drugs and drug delivery vehicles is gaining a lot of attention in recent years. We can conclude from all these observations that for the study of siderophore-inspired molecules, their activity and mechanism of action when culturing mycobacteria in glycerol might be especially well suited.

### 3.6. Investigation of DMEM as Energy Source

A comparison of DMEM vs. glycerol revealed an unexpected metabolic response. Since DMEM contains high amounts of D-glucose, we expected it to, at least partially, copy the metabolic response of the D-glucose vs. glycerol conditions in minimal medium. However, we identified cellular amino acid metabolic processes as the only significant differential expression pathway. From the 15 supplemented amino acids in DMEM, histidine is essentially preferred for nourishing the citric acid cycle. We identified MSMEI_1152 (I7G4Z3), a histidine–ammonia lyase, MSMEI_1148 (A0QRN3, HutU), a urocanate hydratase, as well as MSMEI_1151 (A0QRN6, HutI), an imidazolonepropionase, with all proteins potentially involved in histidine catabolic processes (Figure 7). Our data support that histidine is a substrate of the cytosolic histidine–ammonia lyase MSMEI_1152, which catalyzes the first reaction in histidine catabolism, the nonoxidative deamination of L-histidine to trans-urocanic acid. Subsequently, MSMEI_1148 and MSMEI_1151 further degrade trans-urocanic acid to yield first glutamate and then formate.

Furthermore, we analyzed the presence of proteins associated with *myo*-inositol metabolism, since this molecule is present in DMEM and a potential carbon source (Appendix A). Regarding carbon source utilization, it is striking that in the presence of 1000-times D-glucose, *myo*-inositol is still catabolized by the bacteria, most likely to acetyl-CoA. Unlike diauxic microorganisms, mycobacteria are able to co-utilize carbon sources [38,39]. We identified MSMEI_4532 (I7GCM4) and MSMEI_4547 (A0R191, IolG), two orthologues of *myo*-inositol 2-dehydrogenases, and MSMEI_4544 (I7FQI5, IolB), the *myo*-inositol catabolism IolB protein. IolG is the first enzyme in the *myo*-inositol degradation pathway to 2-keto-*myo*-inositol, which is then subsequently isomerized to 5-deoxy-D-glucuronic acid (Figure 8 and Appendix A). Furthermore, two proteins, MSMEI_4538 (I7G5R0), a sugar ABC-transporter ATP-binding protein, and MSMEI_4540 (I7GDW5, EctC), an ABC-type transporter and periplasmic component, were significantly upregulated by a 3 log fold change. Both proteins are found in the same gene cluster as the identified *myo*-inositol-processing proteins and are part of a predicted ABC transporter including MSMEI_4538 (ATP-binding protein), MSMEI_4539 (ABC transporter sugar permease), and the solute-binding lipoprotein MSMEI_4540. Although these proteins are predicted to be either sugar-binding proteins or xylose-binding proteins, we propose that MSMEI_4538-4540 is a functional inositol type 1 ABC transporter and required for the import of *myo*-inositol into the cytosol. *Myo*-inositol-binding proteins are homologous to xylose- and ribose-binding proteins, since they share common structural motives, and often bind ribose and xylose with low affinity [40]. In order to find supporting evidence, we submitted the protein sequence of the solute-binding protein, which commonly determines the substrate specificity of a given transporter, to Swiss-Model for template identification and model building. Indeed, the template with the highest confidence score of 0.85 ± 0.05 (QMEANDisCo Global) was the monosaccharide ABC-transporter substrate-binding protein ACEI_1806 (PDB: 4ru1) from *Acidothermus cellulolyticus* 11B, a verified *myo*-inositol transporter.

### 3.7. Investigation of D-Glucose as Energy Source

The PCA plot demonstrates that D-glucose and glycerol metabolism exhibit less variability (Figure 3B). D-glucose feeds into the central carbon metabolism upstream of glycerol. Upon comparing the two conditions, we observed multiple downregulated pathways in the presence of D-glucose, with five pathways showing upregulation. Proteins involved in the cellular stress response and peptide metabolic processes were less abundant under the influence of D-glucose. Additionally, alditol, alcohol, and glycerol metabolic processes were downregulated, indicating D-glucose’s involvement in central metabolic pathways related to nucleotide, amino acid, and fatty acid synthesis. Consistent with expectations, cellular carbohydrate metabolic processes were also downregulated; several proteins involved in glycerol metabolism are less produced in the absence of glycerol.

A notable finding was the upregulation of transport pathways, particularly those involving transmembrane transporters, when comparing glucose to glycerol conditions. Several transport proteins displayed significant changes in abundance, including MSMEI_6584 (I7FP00, four-fold log_FC_), an ABC-2-type transport system ATP-binding protein, MSMEI_4110 (I7G4L4, three-fold log_FC_), a putative tricarboxylic transport membrane protein, and MSMEI_1642 (I7G662, three-fold log_FC_), a permease for cytosine/purines uracil thiamine allantoin. However, intriguingly, the putative glucose permease (A0QZX8, MSMEI_4089), previously predicted by Titgemeyer and colleagues in 2009, did not exhibit differential expression in our dataset, challenging the previous assumption [41].

Based on the composition of the growth medium, there is a possibility that the putative tricarboxylic acid transporter MSMEI_4110 may be involved in the transport and uptake of citric acid. Previous studies conducted in our laboratory revealed that *M. smegmatis* cannot utilize citric acid as its sole energy source at the applied concentration in minimal medium [42]. However, it is plausible that in the presence of D-glucose, the citric acid cycle is activated, leading to the active consumption of citric acid and the induction of relevant transport pathways. Yet, the reason for the higher activity of this transport pathway with D-glucose compared to glycerol remains unclear. Notably, the upregulation of MSMEI_4110 exhibits a similar pattern (two-fold log_FC_) when the D-glucose condition is compared with all other applied conditions.

Furthermore, we have not reached a definitive conclusion regarding the roles of the two transporters, MSMEI_6584 and MSMEI_1642, in nutrient uptake. Considering the crucial significance of membrane transport proteins in bacterial survival and adaptation, it is imperative to conduct further investigations to experimentally verify their dependence on glucose supply. These identified transmembrane proteins warrant thorough follow-up studies to gain deeper insights into their functions and regulatory mechanisms. Such validations will contribute significantly to our understanding of bacterial physiology and potential drug targets.

### 3.8. Investigation of Tween 80 as Energy Source

Tween 80 (Polysorbate 80) is a polysorbate-type nonionic surfactant formed by polyethoxylated sorbitan and oleic acid, yielding a molecule with 20 units of polyethylene glycol, one sorbitol, and one oleic acid ester unit. Differential expression analysis showed that four pathways are upregulated in the presence of Tween 80 vs. glycerol: the sterol catabolic process, the steroid catabolic process, the secondary alcohol metabolic process, the cholesterol metabolic process, and the cholesterol catabolic process. All the identified processes are involved in fatty acid-like metabolism or catabolism; hence, the major driving force of Tween 80-supported growth is potentially the hydrolyzed oleic acid unit that is available as an energy source. An earlier metabolome analysis of *M. tuberculosis* supplied with Tween 80 found a similar footprint with *M. tuberculosis* grown in vivo in a lipid-rich environment [43], and our data support this finding. Oleic acid is a monounsaturated omega-9 fatty acid (18:1, cis-9) and can directly feed into the fatty acid cycle (Figure 9). Several membrane transporters have been implicated in fatty acid uptake, especially from the Mce family, including Mce1 and Mce4 [44]. In our dataset, however, we observed downregulation of Mce family members, and this potentially points towards other uptake pathways of oleic acid derived from Tween 80. Indeed, one of the top upregulated hits (seven-fold log_FC_) is an ABC transporter-related protein (I7FN68, MSMEI_6324) with an unknown function, as well as a periplasmic substrate-binding-like protein (I7GD24, MSMEI_4112, eight-fold log_FC_ upregulation), often involved in solute transport, with an unknown function.

Another interesting observation in this dataset is that processes involved in external encapsulating structure organization and cell wall organization/biogenesis are downregulated in the Tween 80 condition. This especially concerns capsule polysaccharide synthesis proteins. Potentially, these molecules are not required since oleic acid is directly incorporated into the cell envelope. Previous studies in the organism *Lactobacillus plantarum* (*L. plantarum*) reported that Tween 80 supplementation increases the cellular proportions of oleic acid and the direct incorporation of exogenous fatty acids into cellular lipids [45]. The observed higher membrane rigidity of *L. plantarum* resulted in reduced permeability to small molecules, such as toxic compounds and dyes. On the contrary, a study in *Mycobacterium avium* reported an increased activity of several anti-tubercular drugs in the presence of Tween 80 in the culture medium [46]. As a result, we can conclude that the additive Tween 80 in mycobacterial cultures during drug screening efforts is potentially not optimal, since the membrane permeability towards small molecules might be altered. It is also noteworthy that well-known drug targets reside within the membrane (e.g., DprE1–Benzothiazinone s [47,48] or MmpL3–SQ109 [49]) and several of these are downregulated in the Tween 80 condition.

### 3.9. Investigation of Cholesterol as Energy Source

Through our approach, we successfully confirmed the overexpression of several proteins involved in the beta-oxidation of the cholesterol side chain (Figure 10), as well as key enzymes responsible for the degradation of the sterol ring, namely MSMEI_5766 (kshA, A0R4R3) and MSMEI_5878 (kshB, A0R525). This catabolic process predominantly yields propionyl-CoA and acetyl-CoA, with smaller amounts of pyruvate or succinyl-CoA [50].

Propionyl-CoA is a metabolite rich in energy; however, high concentrations can be potentially toxic to the cell [51]. To mitigate this, propionyl-CoA can follow various utilization pathways, including entering the methylcitrate cycle (MCC) where it is metabolized into pyruvate [52,53]. Notably, specific enzymes within the MCC, such as methylcitrate dehydratase MSMEI_6466 (I7GBC1, two-fold log_FC_) and aconitase MSMEI_1081 (I7G321, four-fold log_FC_), were significantly upregulated in our dataset.

Another way to utilize propionyl-CoA is in the biosynthesis of odd-chain fatty acids, which are crucial components of cell wall lipids and play a pivotal role in mycobacterial virulence. Consequently, several long-chain fatty acid-CoA ligases, including MSMEI_5853 (I7GFA1, fadD3), MSMEI_5748 (I7G979, fadD17) and MSMEI_3051 (I7FDC7, fadD11), were upregulated in the presence of cholesterol. These ligases generate fatty acyl-CoAs, which can serve as substrates for both beta- and alpha-oxidation and can be used for the synthesis of complex lipids or to modify proteins [54]. Given the presence of cholesterol as a high-energy source, we believe the latter to be the case in this context.

The primary cholesterol import system in *M. tuberculosis* is believed to be the Mce4 transporter, an ABC-like ATP-dependent cholesterol import system encoded by the *mce4* locus [55,56]. However, our findings present a contrast to previous research, as the homologous proteins of the Mce4-family transporter in *M. smegmatis* were slightly downregulated in the presence of cholesterol, except for Mce4E (MSMEI_5736, I7FLI0, alternative gene name: LprN). Additionally, we could not confirm the necessity of LucA (I7FUI6, MSMEI_6123, downregulated by 1.3 log_FC_) in cholesterol uptake, similar to the observations made by Rank et al. [57]. This discrepancy may be related to the fact that, unlike previous studies, we did not use cholesterol dissolved in a carrier such as tyloxapol:ethanol or methyl-ß-cyclodextrin micelles but instead directly applied it to the growth medium. This difference in methodology might have altered the cholesterol uptake pathway in the cell. For instance, although *M. tuberculosis* cannot use ethanol or the combination of tyloxapol and ethanol to support significant growth, these agents still encapsulate cholesterol molecules, potentially leading to significant changes in transport pathways [56].

Remarkably, our dataset revealed a substantial upregulation of the putative membrane transporter MSMEI_6324 (I7FN68, 5.4 log_FC_). Intriguingly, this transporter exhibited similarly high levels of overproduction under the Tween 80 condition, while its presence remained negligible in all other datasets (Appendix A). This dual occurrence of upregulation suggests a potential link between MSMEI_6324 and lipid metabolism. Furthermore, our analysis identified two ABC-transporter proteins, MSMEI_4110 and MSMEI_4112, both showing significant upregulation. Consistent with MSMEI_6324, these ABC transporters also demonstrated notable overexpression under the Tween 80 condition. The shared pattern of upregulation in both conditions raises intriguing possibilities regarding their role in lipid transport or metabolism.

In summary, our study provides novel insights into the regulation of cholesterol import in *M. smegmatis* and highlights the importance of considering different experimental conditions and methods when investigating complex cellular processes like cholesterol uptake. These findings open new avenues for future research and emphasize the need for additional studies to fully elucidate the mechanisms governing lipid transport in mycobacteria.

### 3.10. Self-Directed Online Comparative Analysis

In order to make our data available to the broader community, we have converted our results to an online tool employing Shiny (https://klemens-froehlich.shinyapps.io/Mycobacterium/) (accessed on 4 December 2023). The application allows the user to screen the different conditions applied in our screening (D-glucose, DMEM, L-lactate, glycerol, Tween 80, and cholesterol) and analyze, as well as visualize, all proteins found in the individual conditions either by individual protein or by bulk search. Visualization is provided as a volcano plot (adj. *p*-value vs. log_FC_), as principal component analysis, and protein abundance. The differential analysis can be visualized for individual growth conditions against both (i) glycerol or (ii) against all other conditions combined. The second option is aimed at providing information on which proteins are solely up- or downregulated in the conditions of interest. Furthermore, whole predicted enrichment terms can be visually inspected. The tool is easily scalable so further screenings, including more conditions, will be added in the future. We invite researchers to explore the data in this low-threshold web application.

## 4. Conclusions

Our dataset represents a meticulously curated collection of high-quality data, showcasing *M. smegmatis* growth under various conditions. Notably, we have deliberately included well-defined carbon sources in this screen, a crucial consideration as even trace amounts of carbon source contaminations can activate metabolic pathways that would otherwise remain inactive. The proteomics analysis conducted on complex media like DMEM corroborates *M. smegmatis*’ ability to efficiently co-utilize energy sources to optimize its growth. This metabolic flexibility is believed to bestow a competitive advantage upon *M. smegmatis*, allowing it to outcompete other microbes within the same ecological niche. Intriguingly, many natural environments host diverse mixtures of nutrients, actively supporting the growth of bacteria. This scenario holds true for intracellular pathogens residing in infected tissues or soil bacteria like *M. smegmatis*, which have access to multiple nutrients from decomposed matter. Our screen serves as a poignant reminder that the development of drugs targeting bacterial central metabolism and energetics is a complex task. The essentiality of these targets may be conditional and influenced by various factors, including the composition of the culture broth medium. Along these lines, we plan to investigate mycobacterial proteoforms in the future. These proteoforms might play a significant role in the adaptability and virulence of mycobacterial species as they can affect protein function, localization, or interactions with host cells. Understanding and characterizing proteoforms is essential for unraveling the complexities of mycobacterial biology. In the era of data management and sharing, our online tool for studying and analyzing the generated data proves invaluable. It not only benefits our research group but also serves as a resource for the broader scientific community. By facilitating access to large datasets, other research groups can build upon our findings and accelerate their own investigations, fostering collaborative progress in the field.

## Figures and Tables

**Figure 1 proteomes-11-00039-f001:**
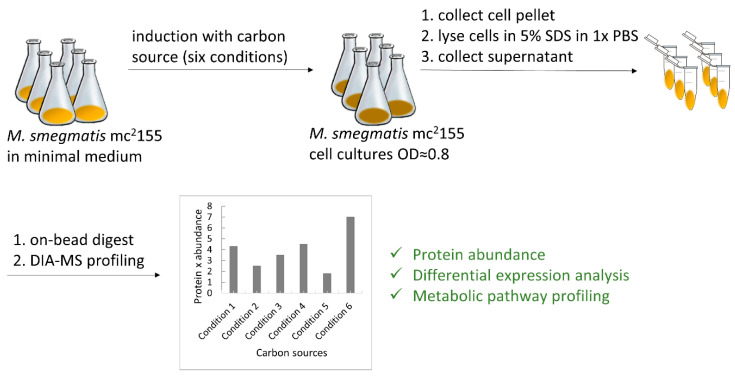
Workflow of experimental setup.

**Figure 2 proteomes-11-00039-f002:**
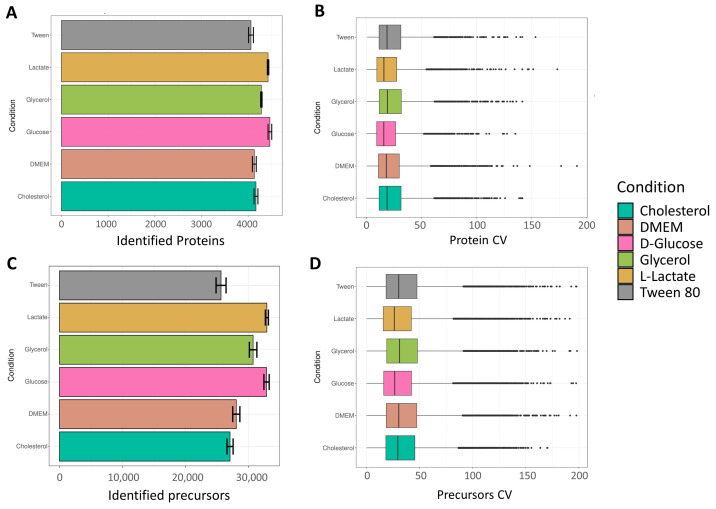
Proteome analysis of *M. smegmatis* grown at six different conditions applying various carbon sources. (**A**) The number of proteins identified in each condition is plotted. (**B**) Boxplots showing distribution of coefficients of variation of non-logarithmic protein groups abundances. (**C**) The number of precursors identified in each condition. (**D**) Boxplots showing distribution of coefficients of variation of precursors.

**Figure 3 proteomes-11-00039-f003:**
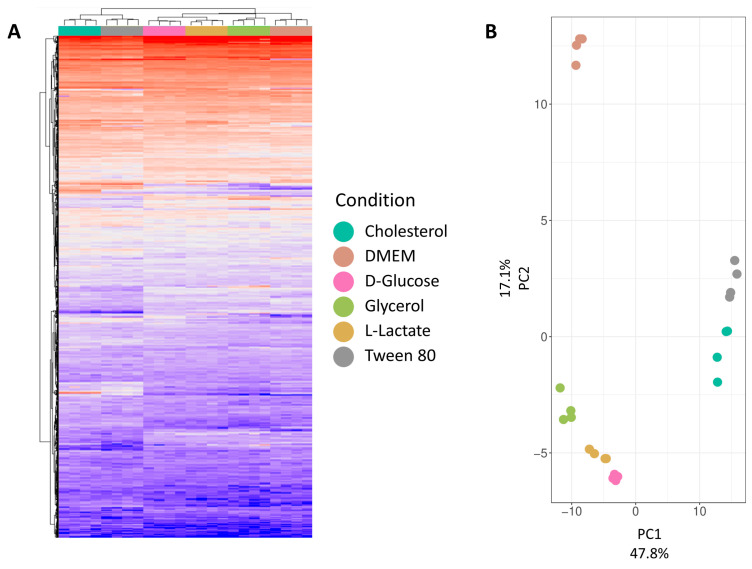
HCA and PCA of total proteomic data between different conditions. (**A**) Hierarchical clustering analysis of all carbon source conditions on protein group level. (**B**) The results of principal component analysis represented as 2-dimensional diagram.

**Figure 4 proteomes-11-00039-f004:**
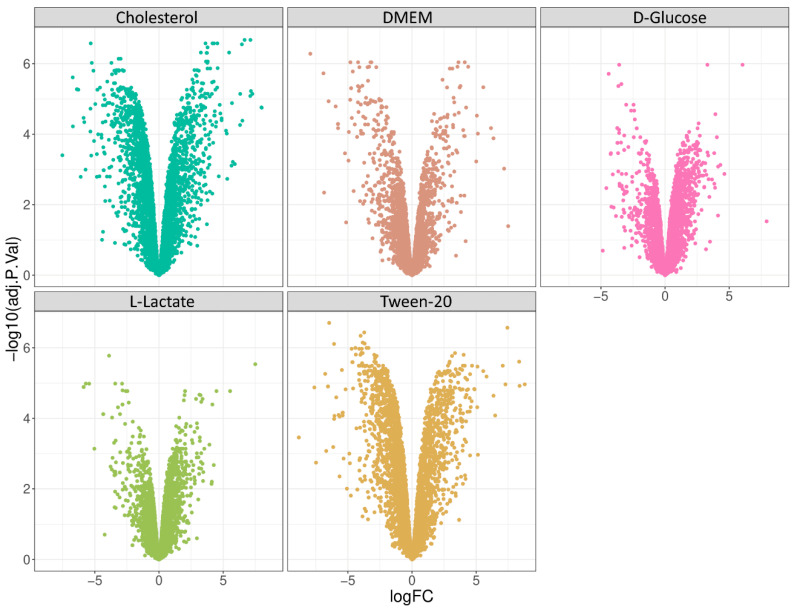
Volcano plot of differential expression analysis of genes in response to the use of different carbon sources for the cultivation of *M. smegmatis* in comparison to glycerol condition.

**Figure 5 proteomes-11-00039-f005:**
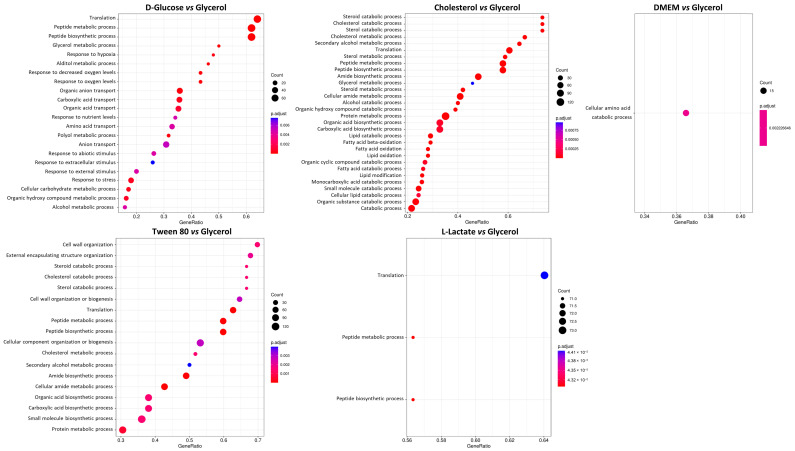
Gene Ontology (GO) enrichment analysis of significantly altered proteins in different carbon source conditions. The *y*-axis shows the GO pathway term. The *x*-axis shows the gene ratio, which equals the number of differentially expressed genes against the number of genes associated with a GO term in wheat genome. The dot color indicates the *p*-value and the area of the dots represents the number of genes enriched in a GO term.

**Figure 6 proteomes-11-00039-f006:**
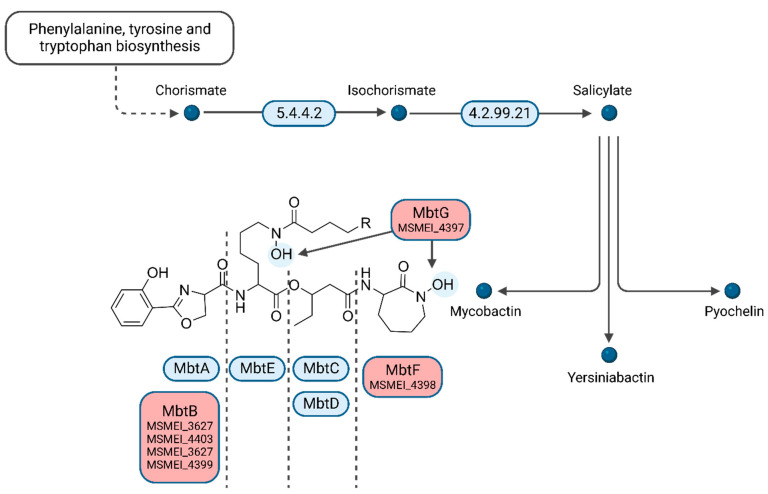
Biosynthesis of siderophore group nonribosomal peptides. Enzymes indicated in red were shown to be down-regulated in the L-lactate condition vs. glycerol (Figure adapted from the KEGG database. Created with BioRender.com) (accessed on 4 December 2023).

**Figure 7 proteomes-11-00039-f007:**
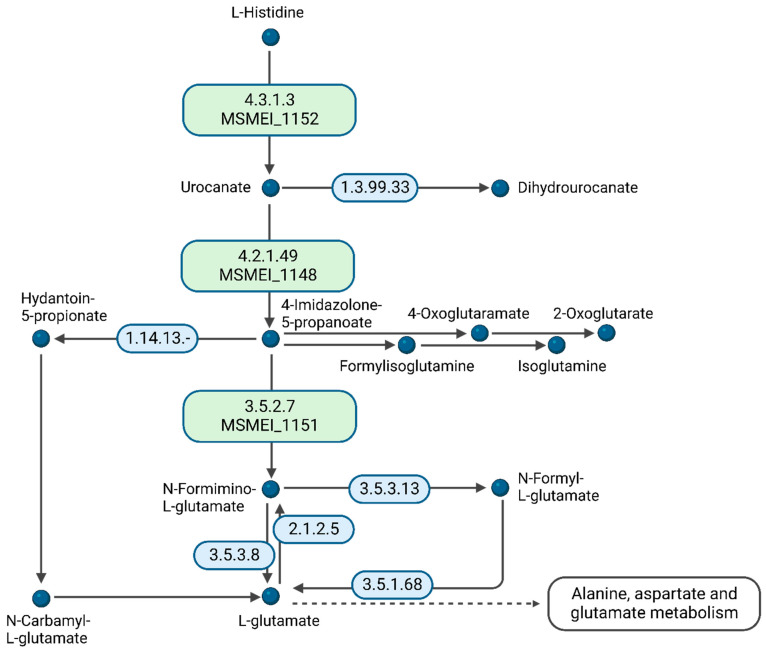
A diagram of histidine catabolism. Enzymes in green indicate the up-regulated proteins in DMEM condition in comparison to glycerol (Figure adapted from the KEGG database. Created with BioRender.com) (accessed on 4 December 2023).

**Figure 8 proteomes-11-00039-f008:**
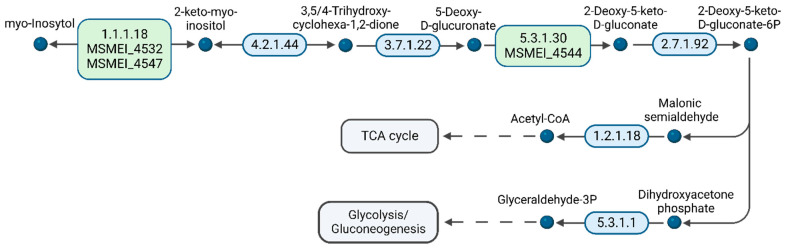
A diagram of *myo*-inositol catabolism. Enzymes in green indicate the up-regulated proteins in DMEM condition in comparison to glycerol (Figure adapted from the KEGG database. Created with BioRender.com) (accessed on 4 December 2023).

**Figure 9 proteomes-11-00039-f009:**
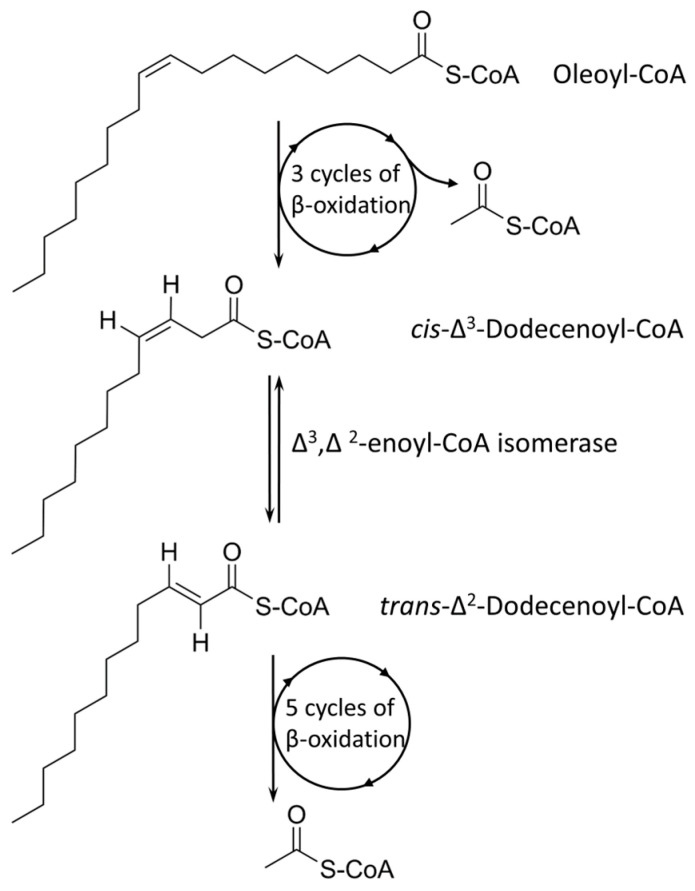
Oxidation of oleic acid.

**Figure 10 proteomes-11-00039-f010:**
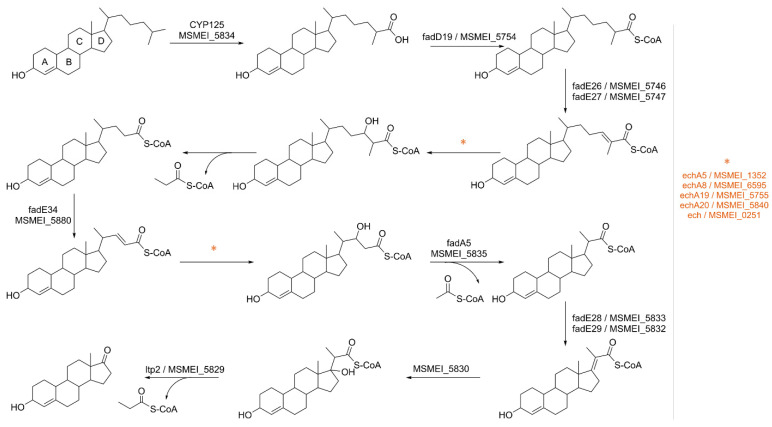
Cholesterol side-chain catabolism in *M. smegmatis.* Confirmed upregulated enzymes are denoted with the enzyme/gene name(s) next to the reaction arrow. Asterisks indicate upregulated possible enzymes for corresponding reactions.

## Data Availability

All raw files, file annotation, FASTA database, DIA-NN output, STRING prediction output, and the R code have been deposited at MASSIVE and is are accessible via the identifier MSV000092025 (password “myco”). The code for the Shiny app is available upon reasonable request from the authors.

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
