# Peer review of "Deep Proteomic Investigation of Metabolic Adaptation in Mycobacteria under Different Growth Conditions"

_proteomes, 2023, doi:10.3390/proteomes11040039_

Round 1
Reviewer 1 Report
Comments and Suggestions for Authors
Zmyslia et al. present a proteomic comparison of M. smegmatis under different metabolic growth conditions in their manuscript.
The manuscript is well written and the figures are well presented. Their online website tool to checkup specific M. smegmatis proteins and their expression pattern under different growth conditions is a great repository for accessing the data, on top.
The figure descriptions of the supplemental material are in need of a proofreading to get rid of remaining spelling mistakes.
The scientific community will be interested in the findings presented in this manuscript.
Please proofread the figure descriptions of the supplemental material.
Author Response
Point-by-Point Response: Manuscript ID “proteomes-2658962”
We would like to thank the reviewers for taking their time as well as for their very helpful suggestions and hope that this improved version of our manuscript will now be appropriate for publication in Proteomes. Please note that changes made are highlighted in yellow (main text and SI).
Please see below our detailed responses to the reviewer questions.
Reviewer 1
Comments and Suggestions for Authors
Zmyslia et al. present a proteomic comparison of M. smegmatis under different metabolic growth conditions in their manuscript. The manuscript is well written and the figures are well presented. Their online website tool to checkup specific M. smegmatis proteins and their expression pattern under different growth conditions is a great repository for accessing the data, on top. The figure descriptions of the supplemental material are in need of a proofreading to get rid of remaining spelling mistakes. The scientific community will be interested in the findings presented in this manuscript.
Comments on the Quality of English Language
Please proofread the figure descriptions of the supplemental material.
We thank reviewer 1 for the overall very positive assessment of our manuscript. We apologize for the spelling mistakes in the supplemental material. We have corrected the mistakes.
Reviewer 2 Report
Comments and Suggestions for Authors
The present from Zmyslia et al. is a comprehensive study to understand the metabolic adaptation of M. smegmatis mc2155 to various carbon sources. This study is very insightful with respect to expanding the knowledge of protein expression profiles to different carbon sources in culture conditions. In a broader sense, it will help utilize the right culture condition to mimic bacterial environment to understand bacterial pathophysiology and for drug screening. The manuscript is well-prepared and easy to understand. The authors However, I have one major critique, even though the authors have analyzed the protein expression profiles for all the conditions and compared and contrasted them, it would be beneficial to compare the proteomic profiles of the culture conditions to the proteomic profiles of Mycobacteria tuberculosis grown in vivo, such as “Kruh NA, Troudt J, Izzo A, Prenni J, Dobos KM (2010) Portrait of a Pathogen: The Mycobacterium tuberculosis Proteome In Vivo. PLoS ONE 5(11): e13938. https://doi.org/10.1371/journal.pone.0013938” or similar studies. This will help elucidate which culture condition may more closely mimic in vivo condition. Also, a comparison with dormancy profiles could be useful though an additional condition of oxygen limitation may provide additional changes, a comparison would increase the utility of the present study for broad purposes.
Major Critiques:
One major experimental issue that I observe is the use of OD600- 0.6-0.8. It is the logarithmic phase for 7H9-grown cultures, but the authors must show evidence that there is similar growth of M. smegmatis under the other conditions tested as well.
Minor Critiques:
In Ln 254, the authors must expand on “non-logarithmic protein group abundances.”
Ln 52-59, the authors must provide adequate references for the examples provided.
Author Response
Point-by-Point Response: Manuscript ID “proteomes-2658962”
We would like to thank the reviewers for taking their time as well as for their very helpful suggestions and hope that this improved version of our manuscript will now be appropriate for publication in Proteomes. Please note that changes made are highlighted in yellow (main text and SI).
Please see below our detailed responses to the reviewer questions.
Reviewer 2
Comments and Suggestions for Authors
The present from Zmyslia et al. is a comprehensive study to understand the metabolic adaptation of M. smegmatis mc2155 to various carbon sources. This study is very insightful with respect to expanding the knowledge of protein expression profiles to different carbon sources in culture conditions. In a broader sense, it will help utilize the right culture condition to mimic bacterial environment to understand bacterial pathophysiology and for drug screening. The manuscript is well-prepared and easy to understand. However, I have one major critique, even though the authors have analyzed the protein expression profiles for all the conditions and compared and contrasted them, it would be beneficial to compare the proteomic profiles of the culture conditions to the proteomic profiles of Mycobacteria tuberculosis grown in vivo, such as “Kruh NA, Troudt J, Izzo A, Prenni J, Dobos KM (2010) Portrait of a Pathogen: The Mycobacterium tuberculosis Proteome In Vivo. PLoS ONE 5(11): e13938. https://doi.org/10.1371/journal.pone.0013938” or similar studies. This will help elucidate which culture condition may more closely mimic in vivo condition. Also, a comparison with dormancy profiles could be useful though an additional condition of oxygen limitation may provide additional changes, a comparison would increase the utility of the present study for broad purposes.
We express our gratitude to the reviewer for the insightful suggestion, which was indeed a point of thorough consideration among the authors prior to submission. After careful deliberation, we reached the consensus that a direct comparison between entirely distinct experimental setups and, notably, quantitation methods would be suboptimal, if not impractical. Consequently, our current approach involves conducting experiments with diverse strains (M. abscessus, M. bovis, and M. tuberculosis, including M. tuberculosis grown in macrophages) within a unified experimental framework. This allows us to investigate and compare these distinct growth conditions in a more cohesive and meaningful manner. We are actively engaged in executing this experimental plan and anticipate reporting the results in due course.
Simultaneously, through our online tool available at https://klemens-froehlich.shinyapps.io/Mycobacterium/, researchers now have the capability to explore specific proteins highlighted in the aforementioned in vivo study. For instance, the top candidate based on spectral counts (SI from https://doi.org/10.1371/journal.pone.0013938), the acetyl carrier protein AcpM, illustrates its abundance in the investigated growth conditions, as depicted in the appended figure.
Since the manuscript on the in vivo proteome provides very interesting information, we have included it in our references.
Major Critiques:
One major experimental issue that I observe is the use of OD600- 0.6-0.8. It is the logarithmic phase for 7H9-grown cultures, but the authors must show evidence that there is similar growth of M. smegmatis under the other conditions tested as well.
We value the constructive feedback provided by the reviewer. In response, our revised manuscript now incorporates supplementary data, specifically growth curves depicted in Supplementary Figure (Figure S1). We have opted to present growth within the OD600 range of 0.6-0.8, as this range corresponds to the logarithmic phase across all experimental conditions.
Minor Critiques:
In Ln 254, the authors must expand on “non-logarithmic protein group abundances.”
In mass spectrometry, intensity data follow log-normal distribution. As many statistics assume a normal distribution, log-transformation has to be performed. However, log-transformation can significantly influence variance estimation. To circumvent this, we calculated the coefficients of variance (CV) of the non-logarithmic protein group and precursor intensities. Please see: DOI: 10.15406/mojpb.2017.06.00200
Ln 52-59, the authors must provide adequate references for the examples provided.
We thank reviewer 2 for the overall positive assessment of our manuscript. In the revised version of the manuscript, we have included the appropriate reference for lines 52-59. 
Reviewer 3 Report
Comments and Suggestions for Authors
Review of Zmyslia et al.
The authors conducted a study to investigate the impact of various carbon sources on mycobacterial growth, using Mycobacterium smegmatis as a model organism. They developed a cost-effective pipeline for proteome analysis to compare the effects of six different carbon sources on Mycobacterial growth.
Additionally, they made their findings accessible through an online tool. The study's findings contribute to the understanding of mycobacterial adaptive physiology and identify potential targets for drug development, which aids in the fight against tuberculosis.
Overall, I found this paper to be informative. The results and discussion are well-written. Although the concept of testing metabolic adaptation by evaluating the response to different carbon sources is not novel, the online data analysis tool provided by this work is beneficial to the scientific community.
While the research provides valuable insights into Mycobacterium smegmatis growth under various carbon source conditions, I would like to address a few points that I think could help the manuscript further.
It should be noted that Mycobacterium smegmatis might not fully represent the pathogenic Mycobacterium tuberculosis.
The overall manuscript seems a bit lengthy.
The manuscript only discusses the impact of carbon sources on mycobacterial growth, without exploring other factors that may influence adaptation.
There is a need for more in-depth study and experimental characterization of the mechanisms governing lipid transport in mycobacteria, as the current understanding is limited.
Line 197 : “Our enrichment analysis is based on predicted terms, since the experimental characterization of M. smegmatis is very limited.” Authors could conduct more experimental characterizations on M. smegmatis.
Author Response
Point-by-Point Response: Manuscript ID “proteomes-2658962”
We would like to thank the reviewers for taking their time as well as for their very helpful suggestions and hope that this improved version of our manuscript will now be appropriate for publication in Proteomes. Please note that changes made are highlighted in yellow (main text and SI).
Please see below our detailed responses to the reviewer questions.
Comments and Suggestions for Authors
The authors conducted a study to investigate the impact of various carbon sources on mycobacterial growth, using Mycobacterium smegmatis as a model organism. They developed a cost-effective pipeline for proteome analysis to compare the effects of six different carbon sources on Mycobacterial growth.
Additionally, they made their findings accessible through an online tool. The study's findings contribute to the understanding of mycobacterial adaptive physiology and identify potential targets for drug development, which aids in the fight against tuberculosis.
Overall, I found this paper to be informative. The results and discussion are well-written. Although the concept of testing metabolic adaptation by evaluating the response to different carbon sources is not novel, the online data analysis tool provided by this work is beneficial to the scientific community.
While the research provides valuable insights into Mycobacterium smegmatis growth under various carbon source conditions, I would like to address a few points that I think could help the manuscript further.
It should be noted that Mycobacterium smegmatis might not fully represent the pathogenic Mycobacterium tuberculosis.
We thank reviewer 3 for the overall very positive assessment of our manuscript. We have added the following sentence (line 94):
While M. smegmatis may not be a comprehensive representation of pathogenic species like M. tuberculosis, the method developed holds promising potential for future application in studying pathogenic species.
The overall manuscript seems a bit lengthy.
We have condensed several sections of the manuscript, aiming for improved clarity.
The manuscript only discusses the impact of carbon sources on mycobacterial growth, without exploring other factors that may influence adaptation.
The primary aim of our analysis is to investigate the impact of carbon sources. However, the method that we describe is applicable to investigate the response to other factors as well. In line 68, we write: “Mycobacteria regulate their metabolism in response to environmental factors that occur during infection, such as acidic pH, actual availability of host-associated carbon sources or defense molecules, oxygen limitation, to name only some examples. Especially the response to the presence of host-associated carbon sources is believed to play a fundamental role in mycobacterial adaptive physiology.” and in line 96: “Our primary interest is in understanding the relative effects of host-associated carbon sources on mycobacteria; the methods used in this study are applicable to the investigation of a wide range of adaptive responses.”
There is a need for more in-depth study and experimental characterization of the mechanisms governing lipid transport in mycobacteria, as the current understanding is limited.
Line 197: “Our enrichment analysis is based on predicted terms, since the experimental characterization of M. smegmatis is very limited.” Authors could conduct more experimental characterizations on M. smegmatis.
We wholeheartedly concur with the reviewer regarding the necessity for a more comprehensive exploration and detailed analysis of mycobacterial lipid transport and metabolism, recognizing its significance in future mycobacterial research. Our analysis aimed to deliver an in-depth proteomics investigation, relying on predicted terms. Moving forward, we intend to leverage the generated information to investigate the functions of the identified proteins through a traditional biochemical approach.
Reviewer 4 Report
Comments and Suggestions for Authors
Overall, the authors have discussed the proteome analysis of Mycobacterium smegmatis under six different carbon source conditions to investigate the impact of various carbon sources on mycobacterial growth. The article is well-written and very detailed. However, there are several minor changes that may be needed:
1. The sections in the results section could be organized more logically and connected for easier understanding. They should also be concise and focused on delivering key points.
2. On Page 4, line 151, it is unclear how the 50% amplitude is defined. Is this specific to the sonicator used? More information is needed, such as the manufacturer and model of the sonicator.
3. The article should specify the amount of protein processed for each sample.
4. The article mentions that the AGC target was set to 3000%, but the accumulation time is 55ms. An explanation is needed for why such a high AGC is necessary.
5. The authors have provided the number of identified proteins, but Figure 2b shows CV% for quantified proteins. It would be valuable to discuss how many proteins were quantified in each condition and how many were quantified without missing values in all conditions.
6. For Figure 4, it is not clear what cutoff was used in the volcano plot to define significantly altered proteins. The article should provide a list of changed proteins in a supplementary table. Additionally, all the proteins discussed in each section of the results should be organized into a single table with detailed information such as fold changes, p-values, etc., to make it easier for readers to follow.
7. There are some typos in the manuscript that require proofreading
Author Response
Point-by-Point Response: Manuscript ID “proteomes-2658962”
We would like to thank the reviewers for taking their time as well as for their very helpful suggestions and hope that this improved version of our manuscript will now be appropriate for publication in Proteomes. Please note that changes made are highlighted in yellow (main text and SI).
Please see below our detailed responses to the reviewer questions.
Comments and Suggestions for Authors
Overall, the authors have discussed the proteome analysis of Mycobacterium smegmatis under six different carbon source conditions to investigate the impact of various carbon sources on mycobacterial growth. The article is well-written and very detailed. However, there are several minor changes that may be needed:
- The sections in the results section could be organized more logically and connected for easier understanding. They should also be concise and focused on delivering key points.
We regret the lack of logical coherence among the sections in the results, as highlighted by reviewer 4. In the revised manuscript, we have condensed several sections of the manuscript, aiming for improved clarity.
- On Page 4, line 151, it is unclear how the 50% amplitude is defined. Is this specific to the sonicator used? More information is needed, such as the manufacturer and model of the sonicator.
The following information (line 156) has been added:
Cell pellets were thawed on ice, resuspended in 0.5 mL of 5% SDS in 1x PBS (w/v) and lysed by sonication on ice using Bandelin Sonopuls HD 2070 ultrasonic homogenisor (Bandelin Electronics, Berlin, Germany, microtip of 3 mm in diameter) for 30 s on/off 2 times at 50% amplitude.
- The article should specify the amount of protein processed for each sample.
We have added the following information (line 162):
Following lysis, 20 µg of cell lysate was subjected to digestion with the SP3 approach using a Freedom Evo 100 liquid handling platform (Tecan Group Ltd., Männedorf, Switzerland).
- The article mentions that the AGC target was set to 3000%, but the accumulation time is 55ms. An explanation is needed for why such a high AGC is necessary.
The AGC target is set to 3000% to allow as many ions as possible to be injected into the Orbitrap. This maximized the dynamic range and increases sensitivity. As 3000% is the maximum recommended AGC target for this machine on MS2 level, we assume there is no negative impact due to charging effects. Furthermore, as the max injection time is set to 55ms there is no negative impact on cycle time. To make our rational clearer to the general readership we have included the following explanation (line 190):
“To maximize the dynamic range and the sensitivity of the method an MS2 AGC target of 3000% was used. Accumulation time was set to 55ms to limit the cycle time. A resolution of 30.000 (at 200 m/z) was used.”
Off note, Thermo defines the AGC targets for MS1 and MS2 differently on the Exploris series (MS1 300% AGC equals the ions of MS2 3000% AGC).
- The authors have provided the number of identified proteins, but Figure 2b shows CV% for quantified proteins. It would be valuable to discuss how many proteins were quantified in each condition and how many were quantified without missing values in all conditions.
We apologize for this mistake. We have found a slight error in the visualization of Figure 2 and the quantified protein groups. We have rectified this error. Additionally, we have included the missing information the reviewer has requested as a new supplementary figure (Figure S2).
- For Figure 4, it is not clear what cutoff was used in the volcano plot to define significantly altered proteins. The article should provide a list of changed proteins in a supplementary table. Additionally, all the proteins discussed in each section of the results should be organized into a single table with detailed information such as fold changes, p-values, etc., to make it easier for readers to follow.
We thank the reviewer for this comment. As we would have to provide tables for each comparison, we deliberately chose to leave it to the reader to use our online tool to check fold changes and q values for each protein. Here, it is easy to visualize every protein mentioned in the manuscript and get a detailed visualization of protein quantities per condition and log2 fold changes and q values for each comparison. We believe that this is a more concise way of presenting the data. Furthermore, all quantities and calculations are deposited at MASSIVE and are freely available via the identifier MSV000092025 (password “myco”).
- There are some typos in the manuscript that require proofreading.
We apologize for the typos in the manuscript and have now corrected the text in the revised version.
Reviewer 5 Report
Comments and Suggestions for Authors
This manuscript describes the proteomic analysis of M. smegmatis cultured in 6 different conditions: Minimal essential medium containing cholesterol, D-glucose, glycerol, L-lactate, or Tween 80, and DMEM. Some specific proteins related to metabolic pathways were identified according to different conditions. Basically, the data presented in this manuscript are not easy to understand for general readers. So, these points should be considered in the description of the manuscript.
My major concern is that I cannot find relevance between the data using M. smegmatis cultured in different carbon source conditions and any condition of M. tuberculosis in microbiologic application fields, although the authors described this issue. M. smegmatis is a fast-growing mycobacteria but M. tuberculosis is slower, which has a significantly different metabolism. In fact, it is a natural fact that bacteria metabolism varies depending on the supply of influencing carbon sources. I think that more valuable or significant data will be obtained if M. smegmatis cultured in the media containing an antimycobacterial drug of sub-MIC concentration is used for analysis. In addition, at least one among the identified target proteins should confirm with classical protein assay such as western blot whether the protein level is increased or decreased to verify that the proteomic method used in this study is reflected in practical conditions.
Minor comment:
The proteins extracted from M. smegmatis cultured in 6 different conditions were analyzed with SDS-PAGE before digestion with the SP3 approach. Please show it as a supplementary figure. This information gives some helpful data.
Author Response
Point-by-Point Response: Manuscript ID “proteomes-2658962”
We would like to thank the reviewers for taking their time as well as for their very helpful suggestions and hope that this improved version of our manuscript will now be appropriate for publication in Proteomes. Please note that changes made are highlighted in yellow (main text and SI).
Please see below our detailed responses to the reviewer questions.
Comments and Suggestions for Authors
This manuscript describes the proteomic analysis of M. smegmatis cultured in 6 different conditions: Minimal essential medium containing cholesterol, D-glucose, glycerol, L-lactate, or Tween 80, and DMEM. Some specific proteins related to metabolic pathways were identified according to different conditions. Basically, the data presented in this manuscript are not easy to understand for general readers. So, these points should be considered in the description of the manuscript.
My major concern is that I cannot find relevance between the data using M. smegmatis cultured in different carbon source conditions and any condition of M. tuberculosis in microbiologic application fields, although the authors described this issue. M. smegmatis is a fast-growing mycobacteria but M. tuberculosis is slower, which has a significantly different metabolism. In fact, it is a natural fact that bacteria metabolism varies depending on the supply of influencing carbon sources. I think that more valuable or significant data will be obtained if M. smegmatis cultured in the media containing an antimycobacterial drug of sub-MIC concentration is used for analysis. In addition, at least one among the identified target proteins should confirm with classical protein assay such as western blot whether the protein level is increased or decreased to verify that the proteomic method used in this study is reflected in practical conditions.
We deeply value the insightful comments and suggestions provided by the reviewer. It is recognized that fast-growing mycobacteria, such as M. smegmatis, and slow-growing bacteria like M. tuberculosis, demonstrate distinct metabolic characteristics. Despite this difference, M. smegmatis has a long-standing history as a model organism for studying mycobacterial physiology, serving as a valuable tool for investigating mycobacterial growth behavior in environments outside of biosafety S3 facilities. Furthermore, with the increasing research focus on non-tuberculous fast-growing mycobacteria, such as M. abscessus, there is a renewed potential for re-evaluating the importance of M. smegmatis in basic research. We have added the following sentence (line 41): Moreover, there has been a steady global increase in the occurrence and fatality rates associated with both fast and slow-growing non-tuberculous mycobacterial (NTM) diseases [2].
Our primary objective was to develop a method and an online tool for the straightforward, cost-effective, and swift identification of variations in protein production under diverse growth conditions, utilizing M. smegmatis as a basis. Our intention is to extend this method to study other mycobacterial strains. In fact, we have already commenced a comparative analysis involving different strains, including M. tuberculosis, the results of which will be detailed in future reports.
We are grateful for the suggestion to validate our proteomic results through Western blot analysis. We carefully examined commercially available antibodies for M. smegmatis proteins, as detailed in the attached table. However, while L-lactate 2-monooxygenase (MSMEI_3870) emerged as a suitable target with a significant log-fold change of 5.54 in the L-lactate condition, a challenge arose giving the waiting time for ordering and the deadline set for submitting the revised manuscript.
Considering the time constraints associated with the revision process, we have adopted an alternative approach to validate our findings. Reverse transcription PCR (RT-PCR) can serve as a complementary method to confirm the expression levels of L-lactate 2-monooxygenase. We have chosen RNA polymerase sigma factor SigA (MSMEI_2690) as a control for normalization. The RT-PCR analysis unveiled significant upregulation of MSMEI_3870 under L-lactate conditions (see Figure S3). We believe this alternative validation method is both expedient and reliable within the given timeframe.
Minor comment:
The proteins extracted from M. smegmatis cultured in 6 different conditions were analyzed with SDS-PAGE before digestion with the SP3 approach. Please show it as a supplementary figure. This information gives some helpful data.
We regret the confusion. To clarify, we did not conduct SDS-PAGE prior to digestion. Subsequently, we have revised this paragraph to incorporate the specific protein quantities utilized for each sample (line 162).
Round 2
Reviewer 2 Report
Comments and Suggestions for Authors
N/A
Reviewer 5 Report
Comments and Suggestions for Authors
The authors have successfully revised my questions.